# Ten-year cardiovascular risk among cancer survivors: The National Health and Nutrition Examination Survey

Xiaochen Zhang[1,2], Meghan Pawlikowski[1], Susan Olivo-Marston[1], Karen Patricia Williams[3], Julie K. Bower[1], Ashley S. Felix[1]*

1 Division of Epidemiology, College of Public Health, The Ohio State University, Columbus, Ohio, United States of America, 2 Division of Population Sciences, The Ohio State University Comprehensive Cancer Center, Columbus, Ohio, United States of America, 3 Martha S. Pitzer Center for Women, Children, and Youth, College of Nursing, The Ohio State University, Columbus, Ohio, United States of America

* Felix.20@osu.edu

**Data Availability Statement:** All data are available from the public NHANES database at https://www.cdc.gov/nchs/nhanes/index.htm The name of the dataset include: demographics data (DEMO),

## Abstract

### Background

Cancer survivors have a higher risk of developing and dying from cardiovascular disease (CVD) compared to the general population. We sought to determine whether 10-year risk of atherosclerotic CVD (ASCVD) is elevated among those with vs. without a cancer history in a nationally representative U.S. sample.

### Methods

Participants aged 40–79 years with no CVD history were included from the 2007–2016 National Health and Nutrition Examination Survey. Cancer history was self-reported and 10-year risk of ASCVD was estimated using Pooled Cohort Equations. We used logistic regression to estimate associations between cancer history and odds of elevated ($\geq$7.5%) vs. low (<7.5%) 10-year ASCVD risk. An interaction between age and cancer history was examined.

### Results

A total of 15,095 participants were included (mean age = 55.2 years) with 12.3% (n = 1,604) reporting a cancer history. Individuals with vs. without a cancer history had increased odds of elevated 10-year ASCVD risk (OR = 3.42, 95% CI: 2.51–4.66). Specifically, those with bladder/kidney, prostate, colorectal, lung, melanoma, or testicular cancer had a 2.72–10.47 higher odds of elevated 10-year ASCVD risk. Additionally, age was an effect modifier: a cancer history was associated with 1.24 (95% CI: 1.19–4.21) times higher odds of elevated 10-year ASCVD risk among those aged 60–69, but not with other age groups.

### Conclusions

Adults with a history of self-reported cancer had higher 10-year ASCVD risk. ASCVD risk assessment and clinical surveillance of cardiovascular health following a cancer diagnosis

dietary data (DR1IFF, DR2IFF), examination data (BPX, BMX), laboratory data (TCHOL, GHB, UCPREG), and questionnaires data (DIQ, INQ, MCQ, PAQ, SMQ, ALQ, DPQ) from NHANES 2007-08, 2009-10, 2011-12, 2013-14, and 2015-16.

**Funding:** This work was supported by the National Cancer Institute (F99CA25374501 to XZ and K01CA21845701A1 to ASF). The funders had no role in study design, data collection and analysis, decision to publish, or preparation of the manuscript.

**Competing interests:** The authors have declared that no competing interests exist.

could potentially reduce disease burden and prolong survival, especially for patients with specific cancers and high ASCVD risk.

## Introduction

Despite declining cancer incidence rates among men and stable cancer incidence rates among women [1], the number of cancer survivors continues to increase in the United States (U.S.). Currently, 16.9 million Americans live with a cancer diagnosis [2], with projection models estimating an increase to 26 million U.S. cancer survivors by 2040 [3]. These changes are due in large part to progress in the development of more effective cancer treatments. A major concern for the growing population of cancer survivors, many of whom will live more than 10 years beyond their cancer diagnosis [4], is the occurrence of cardiovascular disease (CVD), such as myocardial infarction, stroke, heart failure, and other heart diseases.

Cancer survivors experience a higher risk of developing and dying from CVD compared to the general population, likely due to multiple mechanisms, including receipt of cardiotoxic cancer treatments [5, 6] and shared risk factors that predispose cancer survivors to subsequent CVD [7]. A recent study from the United Kingdom Clinical Practice Research Datalink [8], which included more than 100,000 cancer survivors and 500,000 individuals without a cancer diagnosis, revealed increased risk of several CVD outcomes, including heart failure, cardiomyopathy, coronary artery disease, and stroke among survivors of certain malignancies as compared to individuals without cancer. Although detailed information on CVD risk factors and cancer treatments were unavailable in this large registry, these findings are consistent with other smaller studies conducted in a Kaiser Permanente cohort [9] and those conducted for site-specific cancers [10], where adjustment for potential confounders was implemented. Together, these studies demonstrate that cancer survivors are more likely to develop CVD compared to individuals without cancer.

In 2013, the American College of Cardiology/American Heart Association introduced the Pooled Cohort Equations to assess the 10-year risk of developing atherosclerotic cardiovascular disease (ASCVD) [11]. The 10-year risk was defined as the risk of developing a first hard ASVCD event, including first occurrence of nonfatal myocardial infarction, nonfatal stroke, fatal coronary heart disease, or fatal stroke, with no history of ASCVD. Given the higher burden of CVD among cancer survivors, estimating future risk of developing ASCVD event is critical to develop suitable interventions for CVD prevention and early detection in this population. Using Pooled Cohort Equations could identify cancer patients who are at risk for future ASCVD events, elicit provider-patient discussions about lifestyle modifications, and prompt use of primary prevention interventions and regular screening for early detection. Understanding the likelihood of future ASCVD events is an important precursor and teachable moment for cancer patients to engage in lifestyle modifications to improve cardiovascular health. However, to date, there is no study comparing ASCVD risk between individuals with and without history of cancer using Pooled Cohort Equations; therefore, we undertook the current analysis to describe 10-year ASCVD risk among those with and without a cancer history, overall, according to cancer-site, and according to age in a sample representative of the U.S. population.

## Materials and methods

### Study population

We used data from the National Health and Nutrition Examination Survey (NHANES) 2007–2016. NHANES is a cross-sectional, stratified, multistage survey of the U.S. population, with

oversampling of underrepresented population subgroups. The National Center for Health Statistics Research Ethics Review Board approved and documented informed consent from all participants [12]. All data were completely anonymized and de-identified before access and analysis.

The current study included adults aged 40–79 years old without a history of CVD. The presence of CVD was self-reported and based on positive affirmation of any of the following conditions: "Has a doctor or other health professional ever told you that you had congestive heart failure/coronary heart disease/angina/heart attack (also called myocardial infarction)/stroke?" Participants were considered to have a positive history of cancer if they answered "yes" to the question "Have you ever been told by a doctor or other health professional that you had cancer or a malignancy of any kind?" Participants who refused or had missing data on cancer status were not included in this analysis. Type of cancer was self-reported and categorized as breast, bladder or kidney, prostate, colorectal, other gastrointestinal (GI, including esophagus, gallbladder, liver, pancreas, and stomach), cervical, ovary, uterus, lung, melanoma, hematologic, thyroid, testis, and other (including bone, brain, nervous system, soft tissue, and more than three cancers).

### Data collection

Demographic characteristics including age, sex, race, marital status, education, and income to poverty ratio were self-reported using a standardized questionnaire. BMI was calculated by study technicians using standard measures of height (meters) and weight (kg), and we classified BMI <25.0 kg/m$^2$, 25.0–29.9 kg/m$^2$, and ≥30 kg/m$^2$ as underweight/normal weight, overweight, and obese, respectively. Daily activities and leisure time activities were measured based on the Global Physical Activity Questionnaire. Physical activity was quantified using the self-reported frequency of vigorous and moderate recreation activities (at least 10 minutes continuously) in a typical week. Participants who reported 0, 1–4, or ≥5 days per week of physical activity were classified as sedentary, physically inactive, and physically active, respectively [13, 14]. A dietary interview was conducted to measure detailed dietary intake information for each participant. Dietary intake was assessed according to the Life's Simple 7 Healthy Diet metric [15, 16]. Specifically, dietary intake in five components was evaluated: ≥4.5 cups fruit/vegetable per day; ≥ three 1-oz whole grain per day; <1,500 mg of sodium per day; ≥ two 3.5-oz servings of fish per week; and <450 calories from sugared drinks per week. Participants with 0–1 components or 2–5 components were classified as poor diet or intermediate/ideal diet, respectively, due to few participants meeting ideal diet criteria. A depression score was calculated using the nine-item Patient Health Questionnaire (PHQ-9) to determine the frequency of depressive symptoms over the past two weeks [17].

### Estimating 10-year ASCVD risk

We implemented the Pooled Cohort Equations (http://tools.acc.org/ASCVD-Risk-Estimator/) to estimate 10-year ASCVD risk [11]. The total score was calculated based on participants' age at completing NHANES survey, high-density lipoprotein cholesterol (HDL-C), total cholesterol (TC), systolic blood pressure, smoking status, and diabetes, stratified by gender and race [11]. HDL-C, TC, and systolic blood pressure were measured by study technicians during the physical examination [12]. Diabetes was determined based on self-reported medical conditions or medication use for diabetes. Smoking status was defined as current vs. not current smoker (including both former and never smokers). Participants with a PCE ≥7.5% were considered to have elevated 10-year ASCVD risk while PCE <7.5% was considered as low 10-year ASCVD risk, consistent with prior literature [11].

## Statistical analysis

All analyses incorporated the NHANES sample weights and accounted for the complex sample survey design using standard methods [18]. Continuous variables were presented as weighted means ± standard error, and categorical variables were presented as weighted frequencies. We used ANOVA and chi-square tests to compare continuous and categorical variables by cancer status (no cancer history vs. positive cancer history), respectively. Unconditional logistic regression was used to estimates unadjusted and adjusted odds ratios (ORs) and 95% confidence intervals (CIs) for the association of cancer status and elevated 10-year ASCVD risk. Adjusted models included BMI, marital status, education level, and income poverty ratio, physical activity, dietary intake, and depression to control for potential confounding. Unconditional logistic regression was used to compare each site-specific cancer to no cancer history in relation to elevated (vs. low) 10-year ASCVD risk. For breast, cervical, ovarian, and uterine cancer, we estimated the OR comparing female participants with no history of cancer. Similarly, for prostate and testicular cancer, we estimated ORs comparing male participants with no history of cancer.

To examine whether the association of cancer status and elevated 10-year ASCVD risk differed by age group, a stratified analysis was conducted by age group (age 40–59 vs. age 60–79). The interaction between cancer status and age group was included in each model and tested using the adjusted Wald test. Since CVD and cancer share certain risk factors and we lacked information on the timing of the CVD diagnosis, additional sensitivity analyses were conducted to include participants with a history of CVD to reduce the potential for selection bias. Statistical significance for the interaction was evaluated as P<0.10 and for all other analyses was P<0.05 [19]. All statistical analyses were completed using Stata MP Version 16.1 (StataCorp, College Station, TX) in December 2020.

## Results

We identified 15,095 adults aged 40 to 79 years with no CVD history and non-missing information regarding cancer status and risk factors used to calculate the Pooled Cohort Equations risk estimate. Weighted mean age at the NHANES examination was 55.2 years old, 53.0% were females, 71.7% were non-Hispanic White, 9.9% were non-Hispanic Black, and 18.4% self-reported other race (including Mexican American, Other Hispanic, Asian, Other or multi-racial). About 70.3% of the population was married, 32.4% had a college or higher education, and 18.3% had <138% income to poverty ratio (Table 1).

In our study population, 13,491 (87.7%) participants self-reported no cancer history while 1,604 (12.3%) self-reported a positive cancer history. Mean time since cancer diagnosis was 11.3 years, with 29.6%, 25.4%, and 45.1% reporting a cancer diagnosis within 5, 5–9.9, and greater than 10 years at baseline, respectively. Compared to those with no cancer history, participants with a cancer history were older (61.8 ± 0.4 vs. 54.3 ± 0.1 years, P<0.001), more likely to be Non-Hispanic White (87.2% vs. 69.5%, P<0.001), have a college or higher education (39.2% vs. 31.4%, P<0.001), less likely to be widowed or divorced (24.1% vs. 21.9%, P = 0.008), and more likely to have an income to poverty ratio >400% (50.8% vs. 43.4%, P<0.001). Interestingly, participants with a cancer history were less likely to report a poor diet compared to participants who never had cancer (65.1% vs. 70.7%, P<0.001) (Table 1).

In our study population, 24.8% of participants were classified as having elevated 10-year ASCVD risk (PCE ≥7.5%). Among those with a cancer history, 35.1% as compared with 23.4% of individuals with no cancer history were classified as having elevated 10-year ASCVD risk (Table 2). Mean estimated 10-year ASCVD risk for individuals with vs. individuals without a cancer history was 8.3 ± 0.4% and 5.1 ± 0.1%, respectively. Comparing individual

**Table 1. Baseline characteristics of 1,604 individuals with a cancer history and 13,491 individuals without a cancer history, 2007–2016 NHANES.**

| | Total | No cancer history | Positive cancer history | P |
|---|---|---|---|---|
| | N = 15,095[1] | n = 13,491 (87.7%)[1] | n = 1,604 (12.3%)[1] | |
| Age, years | 55.19±0.14 | 54.26±0.13 | 61.82±0.44 | <0.001 |
| Gender, % | | | | 0.220 |
| Male | 46.98% | 47.27% | 44.98% | |
| Female | 53.02% | 52.73% | 55.02% | |
| Race, % | | | | <0.001 |
| NH White | 71.66% | 69.49% | 87.20% | |
| NH Black | 9.94% | 10.67% | 4.77% | |
| Other | 18.39% | 19.84% | 8.03% | |
| Marital Status, % | | | | 0.008 |
| Married | 70.26% | 70.18% | 70.84% | |
| Widowed/divorced | 22.15% | 21.88% | 24.14% | |
| Never married | 7.59% | 7.95% | 5.02% | |
| Education, % | | | | <0.001 |
| ≤ High school | 37.87% | 39.37% | 27.15% | |
| Some college | 29.77% | 29.23% | 33.65% | |
| ≥College graduate | 32.36% | 31.40% | 39.20% | |
| Income poverty ratio, % | | | | <0.001 |
| <138% | 18.27% | 18.96% | 13.33% | |
| 138–249% | 17.61% | 17.59% | 17.79% | |
| 250–400% | 19.83% | 20.08% | 18.04% | |
| >400% | 44.29% | 43.37% | 50.83% | |
| Depression score | 1.50±0.02 | 1.51±0.02 | 1.49±0.04 | 0.643 |
| BMI, % | | | | 0.417 |
| Normal | 26.06% | 25.79% | 28.03% | |
| Overweight | 35.55% | 35.71% | 34.46% | |
| Obese | 38.38% | 38.51% | 37.51% | |
| Poor Diet, % | | | | <0.001 |
| No | 29.97% | 29.27% | 34.90% | |
| Yes | 70.03% | 70.73% | 65.10% | |
| Physical Activity Level, % | | | | 0.528 |
| Sedentary | 48.44% | 48.67% | 46.82% | |
| Inactive | 17.18% | 17.17% | 17.28% | |
| Active | 34.38% | 34.16% | 35.91% | |

[1] Unweighted sample size, n (%).

All other analyses incorporated the NHANES sample weights

ASCVD risk factors by cancer status revealed older age, higher systolic blood pressure, and personal history of diabetes (all P<0.001) among those with vs. those without a cancer history.

Table 3 shows associations between cancer status and odds of elevated 10-year ASCVD risk. In the unadjusted model, individuals with a cancer history had two-fold increased odds of elevated 10-year ASCVD risk (OR = 3.00, 95% CI: 2.39–3.77) compared to those with no cancer history. After controlling for BMI, marital status, education level, income to poverty ratio, dietary intake, physical activity, and depression score, individuals with a cancer history had 2.4 times increased odds of elevated 10-year ASCVD risk compared to those without a positive cancer history (OR = 3.42, 95% CI: 2.51–4.66).

**Table 2. Distribution of Pooled Cohort Equations and individual risk factors according to cancer status, 2007–2016 NHANES.**

| | Total | No cancer history | Positive cancer history | P |
|---|---|---|---|---|
| | N = 15,095[1] | n = 13,491 (87.7%)[1] | n = 1,604 (12.3%)[1] | |
| *Estimated ASCVD Risk* | | | | <0.001 |
| Pooled Cohort Equations (PCE) | 5.47±0.10 | 5.07±0.10 | 8.30±0.44 | |
| Elevated 10-yr ASCVD Risk[2] | | | | <0.001 |
| No | 75.20% | 76.63% | 64.92% | |
| Yes | 24.80% | 23.37% | 35.08% | |
| *Individual Risk Factors* | | | | |
| Age group | | | | <0.001 |
| 40–49 years | 34.6% | 37.48% | 14.30% | |
| 50–59 years | 32.5% | 33.20% | 27.22% | |
| 60–69 years | 22.0% | 20.64% | 31.80% | |
| 70–79 years | 10.9% | 8.68% | 26.68% | |
| HDL-C (mmol/L) | | | | 0.350 |
| >1.6 | 27.35% | 27.16% | 28.73% | |
| 1.3–1.6 | 25.77% | 26.07% | 23.65% | |
| 1.2–1.29 | 10.67% | 10.75% | 10.06% | |
| 0.9–1.19 | 27.66% | 27.34% | 29.97% | |
| <0.9 | 8.55% | 8.68% | 7.59% | |
| Total Cholesterol | | | | 0.845 |
| <4.1 | 12.96% | 12.81% | 13.98% | |
| 4.1–5.19 | 36.99% | 37.06% | 36.46% | |
| 5.2–6.19 | 33.09% | 33.12% | 32.94% | |
| 6.2–7.2 | 13.15% | 13.13% | 13.26% | |
| >7.2 | 3.81% | 3.88% | 3.36% | |
| Systolic Blood Pressure (mmHg) | | | | 0.002 |
| <120 | 42.60% | 43.56% | 35.77% | |
| 120–129 | 24.50% | 23.88% | 28.90% | |
| 130–139 | 15.98% | 15.80% | 17.28% | |
| 140–149 | 8.84% | 8.68% | 10.02% | |
| 150–159 | 4.14% | 4.16% | 3.96% | |
| 160+ | 3.95% | 3.93% | 4.08% | |
| Smoker | | | | 0.083 |
| No | 81.82% | 81.49% | 84.17% | |
| Yes | 18.18% | 18.51% | 15.83% | |
| Diabetes | | | | <0.001 |
| No | 88.04% | 88.44% | 85.20% | |
| Yes | 11.96% | 11.56% | 14.80% | |

[1] Unweighted sample size, n (%). All other analyses incorporated the NHANES sample weights

[2] Elevated 10-year ASCVD risk was defined as a PCE ≥7.5%

Participants who reported a diagnosis of bladder or kidney, prostate, colorectal, lung, melanoma, testicular, and other cancers, as well as those who reported don't know of their cancer type had increased odds of elevated 10-year ASCVD risk, compared to those with no cancer history (Fig 1). After adjusting for potential confounders, compared to those with no cancer history, participants with a history of testicular cancer had the highest odds of elevated 10-year ASCVD risk (OR = 11.47, 95% CI: 1.13–116.51), followed by prostate (OR = 9.45, 95% CI:

**Table 3. Odds ratios (ORs) and 95% confidence intervals (CIs) for associations between cancer status and elevated vs. low 10-year ASCVD risk based on Pooled Cohort Equations, 2007–2016 NHANES.**

| | Elevated 10-year ASCVD Risk | | |
|---|---|---|---|
| | OR | 95% CI | P value |
| **Unadjusted Model** | | | |
| Positive cancer history vs. no cancer history | 3.00 | 2.39, 3.77 | <0.001 |
| **Adjusted Model** | | | |
| Positive cancer history vs. no cancer history | 3.42 | 2.51, 4.66 | <0.001 |

All analyses incorporated the NHANES sample weights

Adjusted Model controlled for BMI, race, marital status, education level, income to poverty ratio, dietary intake, physical activity, and depression score

4.53–19.73), bladder or kidney (OR = 7.27, 95% CI: 2.58–20.40), melanoma (OR = 5.84, 95% CI: 2.68–12.73), and lung (OR = 5.03, 95% CI: 1.71–14.80) cancer. Compared to those without cancer history, the odds of elevated 10-year ASCVD risk were higher among those who had breast, other G/I, ovarian, and hematologic cancer, without statistical significance.

We observed a significant interaction between age and cancer status in relation to odds of elevated 10-year ASCVD risk, and therefore present ORs stratified by age groups (Table 4). While the odds of elevated 10-year ASCVD risk did not differ among 40–49, 50–59, and 70–79 year-olds, compared to those without cancer diagnosis, those who had a positive cancer diagnosis had increased odds of elevated 10-year ASCVD risk (OR = 2.05, 95% CI = 1.47–2.85) among 60–69 year-olds (P interaction<0.001, Fig 2.).

In the sensitivity analyses that included individuals with a history of CVD, 15,285 (86.6%) participants self-reported no cancer history while 2,006 (13.4%) self-reported a positive cancer history. Mean estimated PCE was 6.2 ± 0.1% overall and 12.9 ± 0.3% for individuals with a history of CVD (S1 Table in S1 File). Among individuals who had CVD, 27.3% were classified as

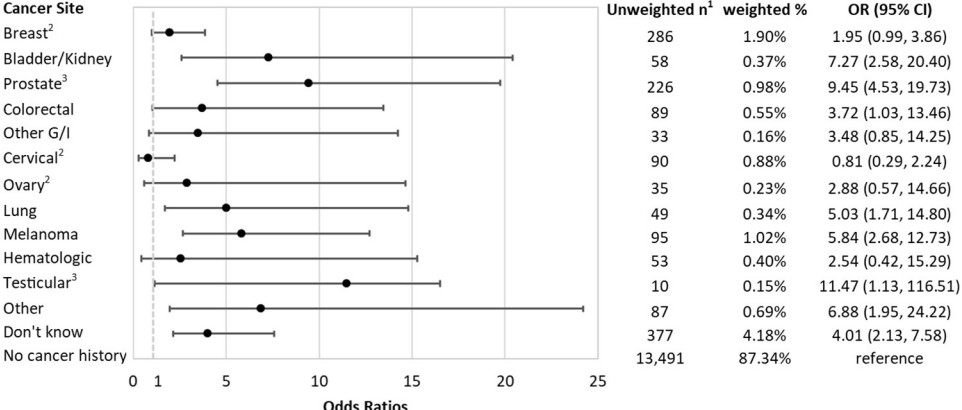

**Fig 1. Odds ratios and 95% confidence intervals of elevated 10-year ASCVD risk (PCEs≥7.5), by cancer sites, compared to those without cancer.** Logistic regression adjsted for BMI, marital status, education level, income to poverty ratio, dietary intake, physical activity, and depression score. [1] Unweighted sample size. All other analyses incorporated the NHANES sample weights. Adjusted Model controlled for BMI, race, marital status, education level, income to poverty ratio, dietary intake, physical activity, and depression score. Other G/I cancer included esophagus, gallbladder, liver, pancreas, and stomach; other cancer included bone, brain, nervous system, soft tissue, more than 3 cancers, and reported as other. [2] with female participants as the comparison. [3] with male participants as the comparison. Effect estimates for cancer in uterus and thyroid were not estimated due to small sample size to calculated population-weighted percentage.

**Table 4. Odds ratios (ORs) and 95% confidence intervals (CIs) for associations between cancer status and elevated vs. low 10-year ASCVD risk based on Pooled Cohort Equations according to age group, 2007–2016 NHANES.**

| Age group | Unweighted sample size, n[1] | weighted % | Elevated 10-year ASCVD Risk | | |
|---|---|---|---|---|---|
| | | | OR | 95% CI | P value |
| 40–49 years | 4,158 | 34.6% | 4.78 | 0.72, 31.91 | 0.105 |
| 50–59 years | 3,716 | 32.5% | 0.47 | 0.20, 1.09 | 0.077 |
| 60–69 years | 3,502 | 22.0% | 2.24 | 1.19, 4.21 | 0.013 |
| 70–79 years | 1,919 | 10.9% | 1.38 | 0.95, 2.01 | 0.088 |

$P_{age\ group\ *\ cancer}$ Interaction $<0.001$

[1] Unweighted sample size, n (%)

All analyses incorporated the NHANES sample weights and adjusted for BMI, marital status, education level, income to poverty ratio, dietary intake, physical activity, and depression score

having elevated 10-year ASCVD risk. The association between cancer status and elevated 10-year ASCVD risk based on Pooled Cohort Equations did not change, and the interaction between age and cancer status in relation to odds of elevated 10-year ASCVD risk remained (S2 Table in S1 File). Compared to the observed association in terms of specific cancer types in the main analysis, we did not observe the higher odds of elevated 10-year ASCVD risk in lung cancer (OR = 2.59, 95% CI: 0.98–6.88), but observed lower odds of elevated 10-year ASCVD risk in thyroid cancer (OR = 0.11, 95% CI: 0.01–0.86) in the model including individuals with a CVD history (S3 Table in S1 File).

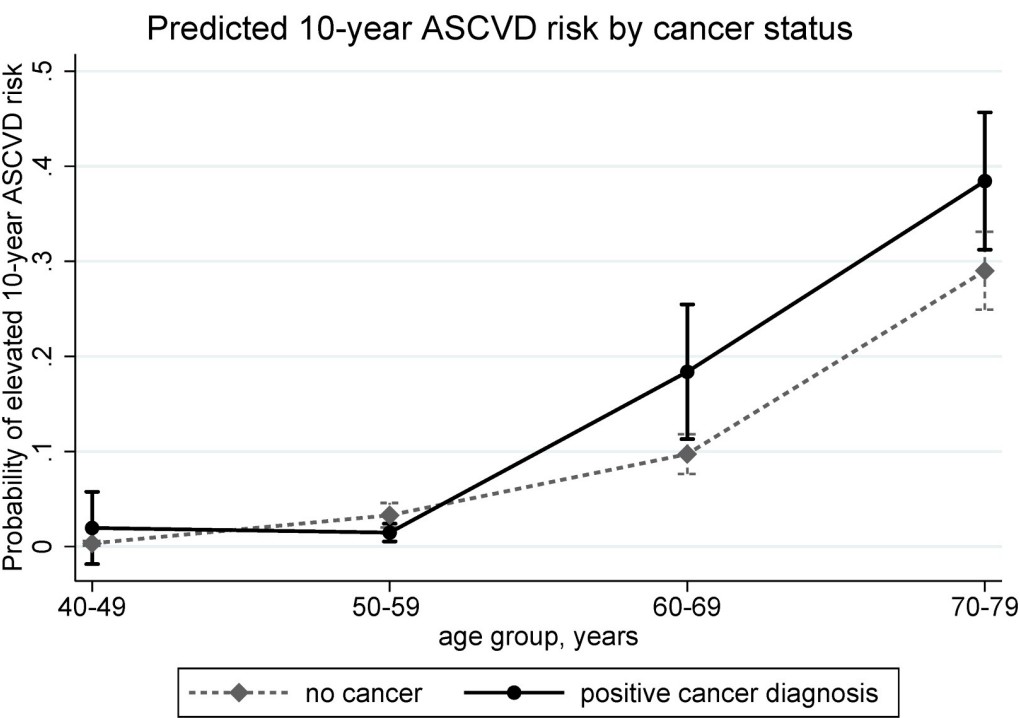

**Fig 2. Estimated population-weighted probability of elevated 10-year ASCVD risk (PCEs≥7.5) between participants with and without cancer diagnosis, according to age groups, based on logistic regression adjusted for BMI, marital status, education level, income to poverty ratio, dietary intake, physical activity, and depression score.**

## Discussion

In this population-based, cross-sectional study, we observed higher estimated 10-year ASCVD risk among individuals with a cancer history compared to those with no cancer history. Specifically, those with a cancer history had two-fold increased odds of elevated 10-year ASCVD risk compared to those with no cancer history. This association varied by cancer type, demonstrating important subgroups who could potentially benefit from CVD prevention interventions. Further, the relationship between cancer history and elevated 10-year ASCVD risk was modified by age, with a statistically significant increased odds of elevated 10-year ASCVD risk observed in the age group of 60–69 year-olds. Although not statistically significant, we observed 3.8 times increased odds of elevated 10-year ASCVD risk for those aged 40–49. Our results have implications for the growing number of U.S. cancer survivors, who are living longer and are at risk of dying from non-cancer-related causes.

Our findings are consistent with prior literature documenting an association of cancer history with increased CVD incidence and mortality, including adult survivors of childhood cancer [10, 20–24]. Increased CVD risks have been observed among breast, lung, prostate, and other cancer survivors from large cohort studies conducted in the United Kingdom, the Netherlands, and the U.S. [8, 9, 25]. Additionally, population-based studies provide some evidence that cancer survivors have a higher burden of subclinical CVD, such as elevated high-sensitivity cardiac troponin T, a marker of subclinical myocardial damage and increased CVD risk, compared to study participants without a cancer history [26].

Our study adds to the current body of literature by examining a composite measure of ASCVD risk—the Pooled Cohort Equations—which has been recommended by the American College of Cardiology/American Heart Association Task Force to predict 10-year risk for first ASCVD event. Few published reports have used the Pooled Cohort Equations to characterize 10-year ASCVD risk among cancer survivors, and no study has compared the 10-year ASCVD risk between those with vs. without a cancer history. In an NHANES study, comparing individuals with cancer history (n = 987) to those without a history of cancer (n = 10,184), elevated 10-year ASCVD was associated with 71% increased risk of cancer-specific mortality [27]. However, this study did not report the estimated Pooled Cohort Equations or the percentage of participants with elevated 10-year ASCVD risk according to cancer history. Several studies have utilized the Framingham risk score (FRS) to compare CVD risk between individuals with and without a cancer diagnosis and reported mixed findings [28]. For example, in a study of 1,222 Korean cancer survivors and 5,196 non-cancer controls, cancer survivors had a higher FRS than those without a cancer history, in line with our study. The average 10-year probability of CVD in relation to the cancer type was significantly higher in patients with hepatic, colon, lung, breast, and gastric cancer [29]. On the other hand, FRS was not significantly higher among those with breast [30], testicular [31], childhood cancer [32], or ovarian cancer [33] compared with controls. The current analysis, which demonstrated increased odds of elevated 10-year ASCVD risk among participants with a history of bladder or kidney, prostate, colorectal, lung, melanoma, testicular, and other cancer, suggests that ASCVD risk estimation may be more clinically relevant for individuals diagnosed with specific cancers.

Age was an important modifier of the association between cancer history and 10-year risk of future ASCVD. Increasing age is strongly associated with an increased risk of developing both cancer and CVD [7]. In line with this notion, compared to those with no cancer history, we observed that those with a self-reported cancer diagnosed between the ages of 60 and 69 had significantly increased odds of elevated 10-year ASCVD risk (OR = 2.24), whereas in other age groups, the odds of elevated 10-year ASCVD risk did not differ according to cancer history. However, we observed a greater relative magnitude of ASCVD risk in the younger age

group (40–49 years), which could be explained by low ASCVD risk in the general population of younger adults. However, as adults age, their ASCVD risk increases, resulting in a smaller relative magnitude of difference between older adults with a cancer history vs. those without a cancer history. This aligns with differences in CVD mortality by age groups observed in previous studies. A recent analysis including 3.2 million cancer patients demonstrated CVD mortality in cancer survivors compared with the general population gradually decreased with increasing age at cancer diagnosis [34]. Further, this study observed higher heart disease mortality among younger cancer patients compared with similarly aged individuals in the general population. Similarly, Zaorsky et al. identified 7.5 million cancer patients from the nationally representative data from the Surveillance, Epidemiology, and End Results and found younger age of cancer diagnosis was associated with a higher standardized mortality ratio of stroke [35]. Our findings, which focus on estimated 10-year ASCVD risk as opposed to CVD mortality, demonstrate that the cancer diagnosis adds to the CVD burden, particularly among younger individuals. This could serve as a forewarning to those diagnosed with cancer in their 40's. Understanding the interaction between age of cancer survivors and ASCVD risk can assist in developing CVD prevention interventions for younger populations, particularly those who are at high risk for ASCVD.

Shared etiologic factors underlying CVD and cancer may contribute to the higher 10-year ASCVD risk we observed among those with a cancer history [36–38]. Indeed, older age and diabetes were more common among those with a cancer history compared to those without a history. We also adjusted for additional ASCVD risk factors not included in the Pooled Cohort Equations model, suggesting the cancer diagnosis adds to the 10-year ASCVD risk independent of CVD risk factors. As we were not able to incorporate temporality of the CVD diagnosis in relation to the cancer diagnosis in our analyses (due to lack of information on the timing of CVD diagnosis), we conducted sensitivity analyses that included individuals with a history of CVD to reduce potential selection bias. We observed similar associations of age, cancer status, and elevated 10-year ASCVD risk, which suggests our findings are unbiased.

The other major mechanism linking cancer with subsequent CVD is attributed to use of certain cancer treatments. Radiation directed at the chest may cause the development of coronary artery disease or blockages [39, 40]. Chemotherapeutic agents, such as anthracycline-based regimens for breast cancer [41], androgen deprivation therapy for prostate cancer [42], and immune-checkpoint inhibitors for melanoma, non-small cell lung cancer, and renal cell cancer [43], can induce cardiac toxicities including vascular compromise, cardiac structural problems, and cardiac dysfunction [44–49]. Practice guidelines call for the evaluation and monitoring of CVD risk factors among cancer survivors for early detection and management of long-term toxic effects from cancer treatment [42, 50–53]. Given the increased CVD burden among cancer survivors, new CVD risk calculators that incorporate factors salient to cancer survivors (*e.g.*, use of chemotherapy/radiation, age at cancer diagnosis, etc.), should be devised to fully account for the increased burden in this population.

Limitations of this study include the cross-sectional study design, potential for recall bias, and lack of information of cancer treatment, clinical data (*e.g.* menopausal status), pertinent tumor characteristics, and timing of each comorbidity. Future studies would benefit from adding treatment type to broaden our understanding of the cardiotoxic profile of cancer treatment and allowing for risk-based surveillance and monitoring for CVD. Finally, we had relatively low numbers of individuals with each specific cancer type, limiting statistical power. Apart from the large, nationally representative sample, which strengthens the generalizability of our results, other strengths include adjustment for potential confounders (*e.g.* socioeconomic factors, dietary intake, and physical activity) and use of standardized measures of height and

weight along with laboratory-based values of lipids and total cholesterol, which allowed us to derive accurate estimates of ASCVD risk.

Our findings suggest that a cancer history is positively associated with increased 10-year risk of ASCVD. As the number of cancer survivors continues to grow annually and considering that 65% of cancer survivors will be alive five years after their diagnosis [3], it is vitally important that we adapt our current screening tools to accommodate the excess CVD risk that a cancer diagnosis might contribute.

## Supporting information

**S1 File. Sensitivity analyses that include individuals with a history of CVD.**
(DOCX)

## Author Contributions

**Conceptualization:** Xiaochen Zhang, Meghan Pawlikowski, Julie K. Bower, Ashley S. Felix.

**Data curation:** Xiaochen Zhang, Meghan Pawlikowski.

**Formal analysis:** Xiaochen Zhang, Meghan Pawlikowski.

**Funding acquisition:** Ashley S. Felix.

**Investigation:** Xiaochen Zhang, Meghan Pawlikowski, Susan Olivo-Marston, Karen Patricia Williams, Julie K. Bower, Ashley S. Felix.

**Methodology:** Xiaochen Zhang, Meghan Pawlikowski, Susan Olivo-Marston, Karen Patricia Williams, Julie K. Bower, Ashley S. Felix.

**Resources:** Ashley S. Felix.

**Supervision:** Julie K. Bower, Ashley S. Felix.

**Writing – original draft:** Xiaochen Zhang, Meghan Pawlikowski, Susan Olivo-Marston, Karen Patricia Williams, Julie K. Bower, Ashley S. Felix.

**Writing – review & editing:** Xiaochen Zhang, Susan Olivo-Marston, Karen Patricia Williams, Julie K. Bower, Ashley S. Felix.

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
