## [Decision Letter · Decision Letter 0]

28 Sep 2020

PONE-D-20-25401

Ten-year Cardiovascular Disease Risk among Cancer Survivors: the National Health and Nutrition Examination Survey

PLOS ONE

Dear Dr. Zhang,

Thank you for submitting your manuscript to PLOS ONE. After careful consideration, we feel that it has merit but does not fully meet PLOS ONE’s publication criteria as it currently stands. Therefore, we invite you to submit a revised version of the manuscript that addresses the points raised during the review process.

The CVD risk score and risk threshold as used in your manuscript are not recommended by the current guidelines anymore. Please change into the Pooled Cohort Equations and risk threshold as recommended by the ACC/AHA guidelines for cholesterol lowering / CVD prevention. The coefficients can be found in Goff Jr et al.  2013 Report on the Assessment of Cardiovascular Risk: Full Work Group Report Supplement.Better justify the use of the 3 models and put it into a context of mediation.Consider a figure with 10-yr CVD absolute risks with and without cancer history based on sample-weighted recycle predictions, survival curves are probably not possible.Please pay careful attention to each comment made by the reviewers.

We look forward to receiving your revised manuscript.

Kind regards,

Bart Ferket

Academic Editor

PLOS ONE

Journal Requirements:

Reviewers' comments:

Reviewer's Responses to Questions

**Comments to the Author**

1. Is the manuscript technically sound, and do the data support the conclusions?

Reviewer #1: Yes

Reviewer #2: Yes

2. Has the statistical analysis been performed appropriately and rigorously? 

Reviewer #1: Yes

Reviewer #2: I Don't Know

3. Have the authors made all data underlying the findings in their manuscript fully available?

Reviewer #1: Yes

Reviewer #2: Yes

4. Is the manuscript presented in an intelligible fashion and written in standard English?

Reviewer #1: Yes

Reviewer #2: Yes

5. Review Comments to the Author

Reviewer #1: The authors analyzed data from NHANES to examine CHD risk among cancer survivors.

The authors found a statistically nonsignificant association with ovarian cancer. This should be mentioned in the Results and Discussion sections.

Reviewer #2: The authors use data from the NHANES to estimate the 10 year cardiovascular disease risk among cancer survivors. Specifically, they include participants 30-65 years of age with no CVD history, from 2007-2016, who provide self reported cancer history. Logistic regression was used to estimate association between cancer history and CVD risk, based on Framingham Risk Scores. They state that patients with reported cancer history had a 3 fold higher OR of 10 year CVD risk.

The study presents results of original research and I do not know of others reporting on this from this database. I do not think this has been published elsewhere. The statistics seem valid, though I have comments about he 3 models below. I would modify the conclusions, per below. The writing is clear. The ethics appear to be good.

Major comments:

When the authors say CVD, I think they mean heart disease / MI risk. Can they please clarify this? CVD is an umbrella term for heart disease, stroke, HTN, etc.

I am not sure why the authors are running so many models and discussing the results of each one throughout the results/discussion. I would pick one model (perhaps the one with the most covariates that the authors find to be most important?). Currently, they have 3 models, and there is a permutation of covariates that may be in these (Table 4). The resulting ORs are confusing to interpret. Since the overall HRs in Table 3 are relatively close to one another, it appears that model 2 and 3 are not much different than model 1. The authors should consider providing a web app to help estimate risk of death from heart disease, with clinicians plugging in the covariates.

This paper needs figures. It is hard to follow the estimates from the tables, esp since there are various models.

Cancer history is per patient report. Is this a limitation of the analysis? For example, how many patients report an incorrect cancer? How many forget to report a cancer?

I would disagree that few studies have examined cardiac risk among cancer survivors (per the intro). Could the authors please discuss how this relates to pediatric cancer patients, who typically have a long survival and are likely to die of competing causes? A recent article on this was published in Cancer: PMID 32298481. Also, the authors should discuss how their findings are similar and different than the recent articles on this topic: PMIDs 32332714, 31729378

Could the authors please display survival curves with death from heart disease vs cancer vs other causes? I realize the ORs may be higher vs the general population, but the absolute numbers may be low.

In the intro, I do not agree with the authors that cancer survival rates are improving because of screening. there has not been a trial for breast, prostate, or lung cancer to show this. In fact, they have all shown no change in overall survival.

Other comments:

I would change this statement: “Cancer history is associated with higher 10-year CVD risk” It is not very meaningful as currently written. can the authors write how it is associated?

6. PLOS authors have the option to publish the peer review history of their article (what does this mean?). If published, this will include your full peer review and any attached files.

Reviewer #1: No

Reviewer #2: No

---

## [Author Response · Author response to Decision Letter 0]

17 Dec 2020

Reviewer #1: The authors analyzed data from NHANES to examine CHD risk among cancer survivors.

The authors found a statistically nonsignificant association with ovarian cancer. This should be mentioned in the Results and Discussion sections.

We added a sentence in the result section to reflect the nonsignificant associations in breast, other G/I, ovarian, and hematologic cancer. We do not comment on these cancers in the discussion section to avoid incorrect overinterpretation, since we did not have the power to detect the significant association. We mentioned the small number of individuals with specific cancers as a limitation to detect significant associations.

Reviewer #2: The authors use data from the NHANES to estimate the 10 year cardiovascular disease risk among cancer survivors. Specifically, they include participants 30-65 years of age with no CVD history, from 2007-2016, who provide self-reported cancer history. Logistic regression was used to estimate association between cancer history and CVD risk, based on

Framingham Risk Scores. They state that patients with reported cancer history had a 3 fold higher OR of 10 year CVD risk.

The study presents results of original research and I do not know of others reporting on this from this database. I do not think this has been published elsewhere. The statistics seem valid, though I have comments about he 3 models below. I would modify the conclusions, per below.

The writing is clear. The ethics appear to be good.

Major comments:

When the authors say CVD, I think they mean heart disease / MI risk. Can they please clarify this? CVD is an umbrella term for heart disease, stroke, HTN, etc.

Thank you for the comment. We revised the introduction section to clarify CVD as cardiovascular disease (such as myocardial infarction, stroke, heart failure, and other heart diseases), and ASCVD event as the “first occurrence of nonfatal myocardial infarction, nonfatal stroke, fatal coronary heart disease, or fatal stroke”.

I am not sure why the authors are running so many models and discussing the results of each one throughout the results/discussion. I would pick one model (perhaps the one with the most covariates that the authors find to be most important?). Currently, they have 3 models, and there is a permutation of covariates that may be in these (Table 4). The resulting ORs are confusing to interpret. Since the overall HRs in Table 3 are relatively close to one another, it appears that model 2 and 3 are not much different than model 1. The authors should consider providing a web app to help estimate risk of death from heart disease, with clinicians plugging in the covariates.

We revised the analysis to use the Pooled Cohort Equations and retained the unadjusted and fully adjusted models for all analyses. The ACA/AHA has a web application to calculate 10-year ASCVD risk for individuals age 40-79 available at the following link: http://tools.acc.org/ASCVD-Risk-Estimator-Plus/#!/calculate/estimate/

This paper needs figures. It is hard to follow the estimates from the tables, esp since there are

various models.

We added figure 1 as suggested to show the association between ASCVD risk and cancer history is modified by age groups.

Cancer history is per patient report. Is this a limitation of the analysis? For example, how many patients report an incorrect cancer? How many forget to report a cancer?

This is a valid concern. As with other self-reported surveys, NHANES does not collect data from medical records. Therefore, we are not able to compare the accuracy of the self-reported cancer diagnosis. We mentioned this as a limitation in the discussion.

I would disagree that few studies have examined cardiac risk among cancer survivors (per the

intro). Could the authors please discuss how this relates to pediatric cancer patients, who typically have a long survival and are likely to die of competing causes? A recent article on this was published in Cancer: PMID 32298481. Also, the authors should discuss how their findings are similar and different than the recent articles on this topic: PMIDs 32332714,

31729378

Our prior statement was overly broad; we agree with the reviewer that many studies have examined risk of developing and/or dying from CVD among cancer survivors; yet few have examined differences in the predictive risk calculators (i.e. Pooled Cohort Equation, Framingham Risk Score, etc.). As such, we revised the introduction and discussion sections to specifically refer to risk assessment among cancer survivors. We state the following in the introduction (page #4) “To date, there is no study comparing ASCVD risk between individuals with and without history of cancer using PCEs.”

We did not discuss pediatric cancer patients because those with pediatric cancer (age <20) were not included in this study. In addition, among the 2,768 participants with cancer history from NHANES 2007-2016, only 81 were diagnosed before 20 years old. Among these, only 23 participants were 40-79 years of age at the time of the NHANES survey. After applying survey weights, these participants represent <0.2% of the population. We understand that for pediatric cancer patients, the competing cause of death is a critical issue. However, our study could not make valid inferences for pediatric cancer patients due to the limited sample, and our focus was not mortality or the cause of death. 

We discussed the similarity of findings from PMID 31729378 with our study in the discussion section (page#16). We did not include the PMID 32332714, since it discussed 1) the fatal heart disease occurred since time of cancer diagnosis and 2) compared age<40 vs. age>=40. Our study focused on age, not time since cancer diagnosis. In addition, after re-analyzing the data using the PCEs, we only include participants 40-79 years old. Therefore, our study is not directly comparable with PMID 32332714.

Could the authors please display survival curves with death from heart disease vs cancer vs other causes? I realize the ORs may be higher vs the general population, but the absolute numbers may be low.

We used the cross-sectional NHANES data, which does not have follow-up and mortality data. Therefore, we were not able to perform survival analysis.

In the intro, I do not agree with the authors that cancer survival rates are improving because of screening. there has not been a trial for breast, prostate, or lung cancer to show this. In fact, they have all shown no change in overall survival.

We have revised the introduction (page#3) to state the following: “These changes are due in large part to progress in the development of more effective cancer treatments.”

Other comments:

I would change this statement: “Cancer history is associated with higher 10-year CVD risk” It is not very meaningful as currently written. can the authors write how it is associated?

We revised the conclusion to “Adults with a history of self-reported cancer had higher 10-year ASCVD risk.”

---

## [Decision Letter · Decision Letter 1]

28 Jan 2021

PONE-D-20-25401R1

Ten-year Cardiovascular Risk among Cancer Survivors: the National Health and Nutrition Examination Survey

PLOS ONE

Dear Dr. Zhang,

Thank you for submitting your manuscript to PLOS ONE. After careful consideration, we feel that it has merit but does not fully meet PLOS ONE’s publication criteria as it currently stands. Therefore, we invite you to submit a revised version of the manuscript that addresses the points raised during the review process.

In particular, please ensure to incorporate the remaining concerns of Reviewer #2 about the use of abbreviations, the Discussion section and lack of figures.

We look forward to receiving your revised manuscript.

Kind regards,

Bart Ferket

Academic Editor

PLOS ONE

Reviewers' comments:

Reviewer's Responses to Questions

**Comments to the Author**

1. If the authors have adequately addressed your comments raised in a previous round of review and you feel that this manuscript is now acceptable for publication, you may indicate that here to bypass the “Comments to the Author” section, enter your conflict of interest statement in the “Confidential to Editor” section, and submit your "Accept" recommendation.

Reviewer #2: (No Response)

2. Is the manuscript technically sound, and do the data support the conclusions?

Reviewer #2: No

3. Has the statistical analysis been performed appropriately and rigorously? 

Reviewer #2: I Don't Know

4. Have the authors made all data underlying the findings in their manuscript fully available?

Reviewer #2: Yes

5. Is the manuscript presented in an intelligible fashion and written in standard English?

Reviewer #2: No

6. Review Comments to the Author

Reviewer #2: The article has improved, and I have some minor comments to further improve the work.

The discussion section is much too long at 6 pages. Can the authors condense this?

Figure 1 is helpful, but can the authors provide other figures of their findings? It is difficult to follow what is statistically significant and meaningful.

The number of abbreviations should be decreased. Clinicians are not familiar with terms like PCEs.

The authors should discuss how their findings relate to pediatric cancer patients (in relationship to PMID 32298481), who typically have a long survival and are likely to die of competing causes. Also, the authors should relate their article on the recent SEER analysis on this topic: 32332714. Specifically, in these works, it appears that the likelihood death from heart disease increases after a cancer diagnosis; however, based on Figure 1, this does not seem to be the case. Can the authors better describe the findings from Figure 1? Can the authors discuss the relationship to the literature more?

I would remove this from the conclusions: “Adapting ASCVD risk assessment and acknowledging cardiovascular health following a cancer diagnosis is critical. Oncologists should advise patients with specific cancers of their potential high ASCVD risk and provide lifestyle modifications to reduce disease burden.” The authors may want to talk about this more in the discussion, but it is not what they found from the data.

7. PLOS authors have the option to publish the peer review history of their article (what does this mean?). If published, this will include your full peer review and any attached files.

Reviewer #2: No

---

## [Author Response · Author response to Decision Letter 1]

15 Feb 2021

Reviewer #2: The article has improved, and I have some minor comments to further improve the work.

The discussion section is much too long at 6 pages. Can the authors condense this?

We have significantly shortened the discussion as requested. 

Figure 1 is helpful, but can the authors provide other figures of their findings? It is difficult to follow what is statistically significant and meaningful.

As requested, we added another figure (Figure 1) to graphically show multivariable-adjusted odds ratios for the association of cancer type and high 10-year CVD risk. To minimize the repetitive information, we removed table 4. 

The number of abbreviations should be decreased. Clinicians are not familiar with terms like

PCEs.

Revised as requested. We kept a few abbreviations to make the article easy to follow.

The authors should discuss how their findings relate to pediatric cancer patients (in relationship to PMID 32298481), who typically have a long survival and are likely to die of competing causes. Also, the authors should relate their article on the recent SEER analysis on this topic: 32332714. Specifically, in these works, it appears that the likelihood death from heart disease increases after a cancer diagnosis; however, based on Figure 1, this does not seem to be the case. Can the authors better describe the findings from Figure 1? Can the authors discuss the relationship to the literature more?

Thank you for the comments. However, our paper focuses on the risk of 10-year high CVD risk, not the deaths from CVD. It is recognized cancer treatment increases the risk of CVD (PMID: 29337636, 31899651, 26919165, 30779651). Our work is based on 10-year CVD risk estimation, by cancer types and stratified by age groups. We are not able to compare our findings with research focus on CVD-specific deaths after cancer diagnosis.

I would remove this from the conclusions: “Adapting ASCVD risk assessment and acknowledging cardiovascular health following a cancer diagnosis is critical. Oncologists should advise patients with specific cancers of their potential high ASCVD risk and provide lifestyle modifications to reduce disease burden.” The authors may want to talk about this more in the discussion, but it is not what they found from the data.

We have removed this statement from the conclusions.

---

## [Editor Report · Decision Letter 2]

17 Feb 2021

Ten-year Cardiovascular Risk among Cancer Survivors: the National Health and Nutrition Examination Survey

PONE-D-20-25401R2

Dear Dr. Zhang,

We’re pleased to inform you that your manuscript has been judged scientifically suitable for publication and will be formally accepted for publication once it meets all outstanding technical requirements.

Kind regards,

Bart Ferket

Academic Editor

PLOS ONE
---

## [Editor Report · Acceptance letter]

24 Feb 2021

PONE-D-20-25401R2 

Ten-year Cardiovascular Risk among Cancer Survivors: the National Health and Nutrition Examination Survey 

Dear Dr. Zhang:

I'm pleased to inform you that your manuscript has been deemed suitable for publication in PLOS ONE. Congratulations! Your manuscript is now with our production department. 

Kind regards, 

on behalf of

Dr. Bart Ferket 

Academic Editor

PLOS ONE